# The SARS-CoV-2 Antibodies, Their Diagnostic Utility, and Their Potential for Vaccine Development

**DOI:** 10.3390/vaccines10081346

**Published:** 2022-08-18

**Authors:** Khalid Hajissa, Ali Mussa, Mohmed Isaqali Karobari, Muhammad Adamu Abbas, Ibrahim Khider Ibrahim, Ali A Assiry, Azhar Iqbal, Saad Alhumaid, Abbas Al Mutair, Ali A. Rabaan, Pietro Messina, Giuseppe Alessandro Scardina

**Affiliations:** 1Department of Medical Microbiology & Parasitology, School of Medical Sciences, Universiti Sains Malaysia, Kubang Kerian, Kota Bharu 16150, Kelantan, Malaysia; 2Department of Zoology, Faculty of Science and Technology, Omdurman Islamic University, Omdurman P.O. Box 382, Sudan; 3Department of Haematology, School of Medical Sciences, Universiti Sains Malaysia, Kubang Kerian, Kota Bharu 16150, Kelantan, Malaysia; 4Department of Biology, Faculty of Education, Omdurman Islamic University, Omdurman P.O. Box 382, Sudan; 5Conservative Dentistry Unit, School of Dental Sciences, Universiti Sains Malaysia, Health Campus, Kubang Kerian, Kota Bharu 16150, Kelantan, Malaysia; 6Department of Conservative Dentistry & Endodontics, Saveetha Dental College & Hospitals, Saveetha Institute of Medical and Technical Sciences University, Chennai 600077, Tamil Nadu, India; 7Department of Restorative Dentistry & Endodontics, Faculty of Dentistry, University of Puthisastra, Phnom Penh 12211, Cambodia; 8Department of Medical Microbiology and Parasitology, College of Health Sciences, Bayero University Kano, Kano 3011, Nigeria; 9Department of Haematology, Faculty of Medical Laboratory Sciences, Al Neelain University, Khartoum 11111, Sudan; 10Preventive Dental Science Department, Faculty of Dentistry, Najran University, Najran 55461, Saudi Arabia; 11Department of Restorative Dentistry, College of Dentistry, Jouf University, Sakaka 72345, Saudi Arabia; 12Administration of Pharmaceutical Care, Al-Ahsa Health Cluster, Ministry of Health, Al-Ahsa 31982, Saudi Arabia; 13Research Center, Almoosa Specialist Hospital, Al-Ahsa 36342, Saudi Arabia; 14College of Nursing, Princess Norah Bint Abdulrahman University, Riyadh 11564, Saudi Arabia; 15School of Nursing, Wollongong University, Wollongong, NSW 2522, Australia; 16Nursing Department, Prince Sultan Military College of Health Sciences, Dhahran 33048, Saudi Arabia; 17Molecular Diagnostic Laboratory, Johns Hopkins Aramco Healthcare, Dhahran 31311, Saudi Arabia; 18College of Medicine, Alfaisal University, Riyadh 11533, Saudi Arabia; 19Department of Public Health and Nutrition, The University of Haripur, Haripur 22610, Pakistan; 20Department of Surgical, Oncological and Stomatological Disciplines, University of Palermo, 90133 Palermo, Italy

**Keywords:** COVID-19, SARS-CoV-2, antibodies, testing, serodiagnostic, immunotherapy, vaccine

## Abstract

Antibodies (Abs) are important immune mediators and powerful diagnostic markers in a wide range of infectious diseases. Understanding the humoral immunity or the development of effective antibodies against SARS-CoV-2 is a prerequisite for limiting disease burden in the community and aids in the development of new diagnostic, therapeutic, and vaccination options. Accordingly, the role of antiviral antibodies in the resistance to and diagnosis of SARS-CoV-2 infection was explored. Antibody testing showed the potential in adding important diagnostic value to the routine diagnosis and clinical management of COVID-19. They could also play a critical role in COVID-19 surveillance, allowing for a better understanding of the full scope of the disease. The development of several vaccines and the success of passive immunotherapy suggest that anti-SARS-CoV-2 antibodies have the potential to be used in the treatment and prevention of SARS-CoV-2 infection. In this review, we highlight the role of antibodies in the diagnosis of SARS-CoV-2 infection and provide an update on their protective roles in controlling SARS-CoV-2 infection as well as vaccine development.

## 1. Introduction

The unprecedented and ongoing spread of coronavirus disease 2019 (COVID-19) has caused a global crisis and prompted widespread concerns [1,2]. As of 21 July 2022, there have been a total of 564,126,546 confirmed cases, and approximately 6,371,354 patients have died in 222 countries [3]. The persisting spreading of SARS-CoV-2 has prompted the scientific community to develop effective vaccine candidates and to produce or find potential drugs or passive immune strategies. The scientific efforts have also focused on acquiring rapid and accurate SARS-CoV-2 diagnostic tests, which are critical for developing effective COVID-19 containment strategies. The human immune response remains the most effective mechanism of combating SARS-CoV-2 infection [4]. Despite the fact that both innate and adaptive immunity are important, SARS-CoV-2-specific humoral immunity has proven to be crucial in determining the disease outcome [5]. Understanding the humoral immunity—or the development of antibodies against SARS-CoV-2—is a prerequisite for limiting disease burden in the community and aids in the development of new diagnostic, therapeutic, and vaccination options [5].

The insufficient testing capacity of the real-time reverse transcriptase polymerase chain reaction (qRT-PCR), particularly in low-resource countries, has highlighted the need for an alternative rapid, simple, accurate, and relatively inexpensive diagnostic approach. For diagnostic purposes, anti-SARS-CoV-2 antibodies represent the most easily identifiable targets [6]. As of now, serological tests have been substantially considered for use as a complements or alternatives to qRT-PCR. Thus, a number of SARS-CoV-2 serodiagnostic tests have been developed and assessed. Many of these tests have proven valuable in detecting SARS-CoV-2 antigens and/or antibodies. Antibody tests have the potential to add important diagnostic value to the routine diagnosis and clinical management of COVID-19. They could also play a critical role in SARS-CoV-2 surveillance for understanding the full scope of the disease and to rebuild public confidence.

While the rapid development of many SARS-CoV-2 vaccines is an extraordinary achievement, the continual emergence of new SARS-CoV-2 variants raises additional questions about the capability of the new virus variants to alter the efficacy of the current vaccine candidates. Therefore, data on the antibody dynamics of SARS-CoV-2-infected individuals or on the vaccination-induced immune responses are critical for understanding vaccine protection and durability, as well as for determining whether additional booster doses are required. Indeed, booster doses are being applied to target certain SARS-CoV-2 variants, such as Omicron [7,8]. Furthermore, the study of specific neutralizing antibodies could aid in the discovery of critical SARS-CoV-2 antigenic regions which could be used in therapeutics and vaccine design.

In this review, we highlight the role of antibodies in the diagnosis of SARS-CoV-2 infection and provide an update on their protective roles in immunotherapy and in current vaccine candidates for the virus.

## 2. Key Proteins of SARS-CoV-2

Understanding the behavior of SARS-CoV-2 key proteins is crucial for developing diagnostic tests, vaccines, and therapeutics. The genome of SARS-CoV-2 encodes both structural and nonstructural proteins. Spike (S), nucleocapsid (N), membrane (M), and envelope (E) are the four key structural proteins that have the potential to be targeted by the antibody response (Figure 1) [9]. The choice of antigen is crucially important in utilizing the virus-specific antibodies for detecting SARS-CoV-2 infection and aiding the development of therapeutics and vaccine candidates. Achieving high sensitivity and specificity when developing antibody tests is mainly dependent on the selection of the diagnostic antigens [10]. However, focusing on the proteins required for viral entry would have a significant impact on the development of any therapeutics or vaccination strategies; the E and M proteins have an essential role in viral assembly [11], and the N protein has been proved to be necessary for viral RNA synthesis (Figure 1) [12]. On the other hand, the S protein plays a crucial role in SARS-CoV-2 attachment and entry into the host cells [13]. Coronavirus S and N proteins are the most antigenic targets for the development of serological assays against SARS-CoV-2. The N protein has high immunogenic potential and is abundantly expressed during the viral infection; however, use of the SARS-CoV-2 N protein is expected to generate more cross-reactivity with other coronavirus strains [14]. The S protein contains the most highly conserved domain among the SARS-CoV-2 isolates [15]. Indeed, the S protein is a key target for eliciting neutralizing antibodies, which are believed to be the major protective effectors against SARS-CoV-2 infection. Therefore, it has been identified as the primary antigenic target for SARS-CoV-2 vaccine development.

## 3. Antibody Response against SARS-CoV-2

Similar to all viral infections, virus-specific antibodies are crucial for the recognition of, clearance of, and protection against SARS-CoV-2. Antibody response to SARS-CoV-2 infection first develops against the N protein. However, protective immunity against SARS-CoV-2 infection is mostly dependent on the neutralizing antibody responses that target the virus’s S protein. Studies on the time and durability of the neutralizing antibody production following SARS-CoV-2 infection revealed that patients begin to generate these antibodies by week two, and the majority of them develop neutralizing antibodies by week three. However, in most recovered patients, independent of age or comorbidities, neutralizing antibody titers gradually declined after 5–8 weeks but continued to be detectable for up to eight months [16].

The transition from a seronegative to a seropositive condition or the development of detectable antibodies in a patient’s serum is known as seroconversion [17]. Many studies have analyzed the kinetics of antibody response in COVID-19 patients. Both severe and nonsevere patients have reported stronger total antibodies as well as single Ig classes’ responses (IgM, IgA, and IgG).

## 4. The Role of Antibodies in SARS-CoV-2 Diagnosis

Currently, real-time reverse transcriptase polymerase chain reaction (qRT-PCR) is essential for confirming SARS-CoV-2 infection [18]. It was quickly established and has become the most reliable assay for viral detection [19]. Despite the wide use, high sensitivity, and specificity of the qRT-PCR, whether it is the gold standard for COVID-19 detection remains unclear [20]. Regardless of the current recommendations for using qRT-PCR in the laboratory diagnosis of suspected people with COVID-19, it has a number of limitations. The primary limitations include viral load and optimal sample types [18]. The efficiency of the detection methods depends on collecting sufficient amounts of viral RNA, and the accuracy of the qRT-PCR for COVID-19 is affected by the quality of the sample. Furthermore, missing the time window of SARS-CoV-2 replication can generate false-negative results, and this incident can happen if samples are collected either at early stages or during recovery [21]. The test also requires a specific specimen type that needs appropriate collection, storage, packaging, and transport [22]. In addition, collecting and handling suspected COVID-19 specimens poses a substantial risk to patients and healthcare workers. Hence, establishing adequate safety measures is mandatory [23]. Other potential limitations are the significant variation in the assay’s sensitivity based on the types of clinical specimens, and the false-negative results as a consequence of improper clinical sampling [24].

As the pandemic exacerbates, medical professionals and researchers are looking for additional epidemiological solutions, and the scientific response to this scenario is based on serological investigation [25]. It is comparatively easier to perform, faster, less expensive, and less complicated than molecular detection [26]. Moreover, the use of blood for most serological tests can reduce the exposure risk amongst healthcare workers handling the samples [17].

Many serological platforms for the detection of SARS-CoV-2-specific antibody responses have been developed and assessed. However, many questions and challenges about serodiagnosis remain: What can COVID-19 antibody tests reveal? Who do we test? When do we conduct a test? What should be tested? Why do we need antigen and antibody tests for COVID-19? This review aimed to provide an update on the recent developments in antibody testing and comprehensively address these questions.

### 4.1. COVID-19 Antibody Tests

Unlike molecular techniques, antibody tests (also known as serological assays) rely on the detection of either viral antigenic proteins or diagnostically detectable antibodies which are created during the immune response to SARS-CoV-2 infection [27]. The main advantage of SARS-CoV-2 antigen tests is that the results—detecting viral antigens in throat or nasal swab samples—can be obtained in minutes [28]. For this reason, using antigen tests is practical, particularly for a large number of people. However, the gradual decline in viral load over time may cause difficulty in detecting viral antigens [29]. Instead, measuring the antibodies produced during SARS-CoV-2 infection has shown great potential for the indirect detection of the virus in a larger time window. In addition, antibody testing can play a crucial role in SARS-CoV-2 contact tracing, surveillance, and epidemiological efforts [30].

Many serological assays have been developed to detect SARS-CoV-2-specific IgM, IgG, and IgA, despite the uncertainty about using these different isotypes individually or in combination [31]. However, none of these antibody isotypes have been clearly identified as the optimal option in the scenario of COVID-19, even though the accurate interpretation of serodiagnostic tests currently depends on the type of antibodies being detected. Indeed, many serological platforms have been utilized to measure the presence of SARS-CoV-2 antibodies and/or antigens (Table 1). These platforms generally take the form of chemiluminescence immunoassays (CLIA) [32,33], enzyme-linked immunosorbent assay (ELISA) [21,34], fluorescence immunoassays (FIA), rapid diagnostic tests (RDTs) and neutralization assays [35,36].

### 4.2. Implications of Seroconversion in SARS-CoV-2 Antibody Tests

The time required by the host immune system to develop an antibody response significantly affects the capability of the serological tests to confirm SARS-CoV infection immediately after a person contracts the virus or during the early stage of infection [56]. Understanding the timing of seroconversion is crucial in determining the optimal time points for specimen collection [57]. Accordingly, establishing or planning any diagnostic protocol involving antibody tests should consider seroconversion time because it has a pivotal role in the efficacy of serodiagnostic tests.

Similar to patients with SARS-CoV, the majority of COVID-19 patients seroconvert after seven days of contracting the virus. The median seroconversion times for IgM and IgG were 12 and 14 days, respectively (Figure 2) [29,58,59]. Even though in some patients, IgM and IgG were detected as early as four or five days (Figure 2) [60,61]. By contrast, delayed antibody response was also reported from a previous study and patients were seroconverted within weeks three or six [62]. The natural delay of antibody production may make antibody tests not well suited for early and accurate detection [17]. However, serological tests that detect SARS-CoV-2 antigens have the potential to identify early infection. Of note, the variation in the seroconversion between different assays reflects their sensitivity [63].

Overall, regardless of the seroconversion time, almost all COVID-19 symptomatic patients seroconvert after their infection with the virus. This suggests that utilizing serological tests in COVID-19 diagnosis among symptomatic patients can serve as a complementary testing approach to the qRT-PCR [62].

Despite the fact that a few studies have addressed the issue of antibody production among asymptomatic patients, seroconversion of anti-SARS-CoV-2 antibodies remains poorly understood [11]. This is because asymptomatic and symptomatic patients exhibit different IgG/IgM response kinetics to SARS-CoV-2. Asymptomatic patients may transmit SARS-CoV-2, highlighting the importance of early diagnosis and treatment [64,65].

In addition, better understanding of the association between seroconversion and disease severity is urgently needed. Importantly, the impact of early or late seroconversion on clinical disease severity is unknown [66], even though a high titer of SARS-CoV-2 antibodies is likely associated with intensive viral loads and severe clinical symptoms [56]. All the above necessitate the need to establish antibody detection assays which are expected to have significant implications for COVID-19 management and control.

### 4.3. Antibody Tests at an Individual Level

Given the uncertainties about the accuracy of antibody tests, additional information is required to support recommending serological testing for use as the sole basis to confirm or exclude active SARS-CoV-2 infection [14]. Nonetheless, and despite critical knowledge gaps, the use of antibody tests in the clinical diagnosis of COVID-19 and assessing immunity to the virus is critical [66]. Given the limited availability of molecular testing, particularly in limited-resource countries, it is necessary to prioritize who gets tested. Accordingly, the WHO, the center for disease prevention and control (CDC), and the European center for disease prevention and control (ECDC) have suggested specific criteria for prioritizing testing. Under this scenario, antigen and antibody tests may offer another avenue to be used as an alternative for diagnosing those not prioritized for molecular testing with specific caution. In general, antibody tests may be utilized at an individual level as a complement to molecular testing for investigating patients with negative qRT-PCR results [26,61]. It may be used to monitor and identify asymptomatic infections amongst close contacts [17]. It may help to understand a patient’s clinical findings, particularly in clinically complicated COVID-19 cases, if multisystem inflammatory syndromes are detected [67]. Antibody tests can be further applied to determine who is qualified to donate convalescent plasma and to screen COVID-19 survivors who undergo convalescent plasma treatments to determine whether they develop their own immunity [68]. Thus, antibody tests may eventually be useful in studying the kinetics of SARS-CoV-2 antibody responses [21]. Serological measurements at baseline, as well as during and after immunization trials, are important for evaluating any vaccine candidate. In all situations, serological results must be carefully interpreted. Finally, testing a pair of serum samples within an interval of two weeks can further increase diagnostic accuracy [69].

A single negative serological test may reflect a false-negative result, so it does not exclude SARS-CoV-2 infection [66,70], particularly in highly exposed persons; this could be due to the low antibody concentrations if the test is performed at the beginning of the infection [68,71], the type of a specimen, or due to the decrease in the number of antibodies after the clearance of the infection. In this case, repeating the test is the best advice. The relatively low negative predictive value of many COVID-19 antibody tests indicates the missing data of many acute infections based on seronegative results [26]. Negative results also allow clinicians to suspect other diseases, because COVID-19 symptoms can resemble those of many other diseases.

By contrast, a positive antigen-based detection test is considered very accurate for identifying acute or early infection [72] and can indicate that a person is likely infected with SARS-CoV-2. Nevertheless, the test positivity may be due to cross-reactivity with other infections, including other human coronaviruses [73]. Therefore, a full-panel test, including other CoVs, SARS-CoV-2, bacterial bronchitis, and influenza, is recommended if applicable [74]. Nevertheless, repeating the tests helps confirm the infection. A fourfold increase in antibody titers is suitable for the diagnosis of SARS-CoV-2. Several factors must be considered to better interpret the results and provide clinically meaningful and valuable recommendations to healthcare providers and test recipients: whether the person is symptomatic or asymptomatic at the time of testing; the positive or negative status of IgM and IgG [74]; and the quality of the test devices.

### 4.4. Population Serological Testing

Nationwide antibody surveillance allows for the tracking of large-scale immunogenicity to COVID-19 vaccines as well as the identification of host variables that may influence antibody formation. Data obtained from antibody testing are used to identify and contain the disease. In addition, the wide application of these tests could transform the battle against COVID-19 [75]. Population serological testing is helpful in assessing and monitoring the spread of the virus [76]. To date, only a few surveillance studies have been conducted, and they primarily targeted symptomatic patients or those who have a severe disease [17,77,78]. Such studies cannot reflect the full spectrum of the disease, as mild or asymptomatic infections may be neglected [79]. Therefore, serologic surveillance is important for defining the infection rate in a population [80] and determining the case fatality rate, the cumulative incidence of COVID-19 [81], and the level of herd immunity in a population [82].

Serological testing can help identify who is infected or exposed and who is immune by assuming protective immunity. Accordingly, population-based serological information will be helpful for officials in making decisions about lifting or enforcing any control measures. The potential of this test in determining the accurate number of infected people in a large population has been tested [83,84]. Population seropositivity indicates that the number of people positive for anti-SARS-CoV-2 antibodies is much higher than that of the reported cases [84,85]. Population-based serological surveillance has been carried out around the world, including countries in Europe [86,87], America [88,89], Asia [90,91], and Africa [92,93]. Indeed, healthcare workers are the population most targeted for serological surveillance due to their high risk of SARS-CoV-2 infection, with ELISA being the most commonly used diagnostic tool for detecting anti-SARS-CoV-2 antibodies.

## 5. Antibody Tests and Seroprotection

At this point in the pandemic, evidence on the status of using antibodies as an immunity passport is insufficient, and we have yet to know whether having antibodies against COVID-19 can protect individuals from future infection. If they do, what is the level of antibodies that provides protective immunity? How long will this potential protection last? Nevertheless, studies have revealed that viral load declines as antibodies form and antibody titers increase. However, different patients can have different antibody responses. According to the WHO, no evidence has supported the idea that patients who recover from COVID-19 are protected against reacquiring infection [94]. Despite the accuracy of antibody tests, results cannot be used as a risk-free certificate, and people should not assume that they are protected against reinfection. Another complicating factor is that most antibody tests measure antibodies that bind to SARS-CoV-2 proteins, and not all binding antibodies are effective in blocking or eliminating viral infection. Therefore, using antibody tests for the prediction of expected immune protection has limitations [95].

As scientists continue to review the evidence, they must understand whether recovery from SARS-CoV-2 infection or the presence of antibodies against the virus will provide future immunity, how it will do so, and whether the severity of subsequent infection will decrease. With these uncertainties, all individuals who have positive results for antibodies will be advised to maintain protective measures. Additional information is urgently needed to determine the strength and duration of generated immunity in individuals who have recovered from COVID-19 and whether some of them can still be infectious or become reinfected [96].

### 5.1. Do Further SARS-CoV-2 Waves Affect the Interpretation of Antibody Test Results?

At the beginning of any pandemic, the positive and negative results of antibody testing can easily be used to discriminate between infected and noninfected individuals. However, with time, real challenges will emerge in terms of differentiating between previously infected, currently infected, and recently recovered patients. In many countries around the world, a massive second or third wave of COVID-19 has already started reporting spikes in cases.

Multiple waves of SARS-CoV-2 have been reported in various countries so far. The new waves of the disease have been setting records, particularly in Europe, the U.S., and Asia. Despite the overwhelming demand for testing during the previous waves [97], health experts are continuously emphasizing the importance of testing to prevent any following waves of SARS-CoV-2 infections [98]. As a result, many countries turned to faster, cheaper tests to contain the second wave and avoid the delays and shortages that challenged the efforts to rapidly diagnose and track infected people. Nevertheless, we are still not ready to establish the widespread use of antibody tests or develop an efficient diagnostic algorithm. Therefore, experts, clinical laboratories, research institutions, industry partners, and government agencies should be involved in a general framework to generate or improve antibody tests and be adequately prepared for a third wave.

The tremendous achievement of acquiring an effective SARS-CoV-2 vaccine added another task to the role of antibody tests. How long will the antibodies generated from the vaccination last? Can they be differentiated from the antibodies generated against the real virus during the testing? These questions remain to be answered. The scientific response to this scenario has been to explore differential antigens and antibodies that have great potential for application in antibody testing to get ready for any future wave of SARS-CoV-2 infections.

### 5.2. Antibodies in SARS-CoV-2 Vaccine Development

Immunity against SARS-CoV-2 infection is normally acquired in one of two ways. Contracting the virus typically ends in natural immunity for a certain period, and vaccination is another way to become resistant. The entry of SARS-CoV-2 into the human cell is the first step of the infection and one of the most crucial processes in the virus’ life cycle. As a result, it is a prime target for vaccinations and therapeutics.

The virus enters the cells of the lung, gastrointestinal tract (GI tract), kidneys, liver, heart, and other organs through the binding of the S protein’s receptor-binding domain (RBD) to its target host receptor, the angiotensin-converting enzyme 2 (ACE2), and a host protease known as transmembrane serine protease 2 (TMPRSS2), which facilitates the cleavage of the S glycoprotein, allowing viral access into the host cells [99,100]. Therefore, one of the main goals of SARS-CoV-2 vaccine development is to generate neutralizing antibodies that block virus entry or prevent membrane fusion. Accordingly, the vast majority of the current vaccines have been designed to induce antibodies against the S proteins due to their protective neutralizing activity [101]. 

Despite the fact that the post-vaccination immune response has several components, including innate, humoral, cellular, and cytokine responses, immunological surveillance that measures antibody response is fundamental for assessing the efficacy of all SARS-CoV-2 vaccines. In particular, measuring the level of circulating anti-S-RBD antibodies could provide important information on SARS-CoV-2 acquired immunity [102]. Indeed, a significant correlation was seen between antibody titers and vaccine efficacy, with higher titers corresponding to better vaccine efficacy [103].

In fact, the level and duration of neutralizing antibodies required for long-term immunity is currently unknown. One study demonstrated that health care professionals who received the BNT162b2 vaccination showed a significant reduction in anti-RBD IgG antibodies after six months (7.4-fold drop). In addition, the decline was 2.5 times greater when evaluated after nine months [104]. The dynamics of the antibody response against SARS-CoV-2 pre- and post-vaccination with the different types of SARS-CoV-2 vaccines have been evaluated in several studies [103]. Most of them reported a gradual and substantial decrease in anti-SARS-CoV-2 antibody levels throughout the first six months following vaccination.

### 5.3. Mechanism of Antibody-Mediated Protection

The role of antibodies in resistance to SARS-CoV-2 infection was explored. Understanding the properties and mechanisms by which antibodies provide protection is essential to defining immunity. Infection or vaccination history may have a role in providing protection against the subsequent infection, and several studies have provided evidence for such protective associations [105]. Upon infection, pre-existing antibodies bind to the surface of SARS-CoV-2 virus particles and lead to the neutralization of the viral spike. Neutralizing antibodies are critical for the efficacy of any SARS-CoV-2 vaccine. Over the past two years, SARS-CoV-2-neutralizing antibodies have been developed for preventive or therapeutic uses [106,107,108]. Most of the neutralizing antibodies target the S protein; their neutralization potency and breadth vary according to recognition epitopes. These findings have prompted an intense effort to identify potential immunodominant epitopes that are recognized by broadly neutralizing the antibodies that could be used as templates for SARS-CoV-2 vaccine design.

Beyond neutralization, recent research suggests that antibodies also mediate other effector functions via the recruitment of complement and/or Fc receptors, which are present on all immune cells and represent important contributors to protective immunity against the SARS-CoV-2 virus. These extraneutralizing functions include opsonization for complement activation, phagocytosis, and antibody-dependent cell-mediated cytotoxicity [109].

Antibodies possess two functional domains: Fab and Fc. The Fab portion of an antibody is highly variable and allows the antibody to recognize and bind to widely diverse molecular structures. The Fc portion of the antibody is more highly conserved and performs two basic functions. First, it interacts with the rest of the immune system after an antibody has bound to its target. This interaction can lead to the activation of one of the aforementioned effector mechanisms.

### 5.4. Prevaccination Antibody Screening

Notably, limited-resource countries (LRCs) remain at the back of the line when it comes to new technologies, infrastructure, and public health control measures such as vaccines. Evidently, COVID-19 mass vaccination is currently not applicable in most of the LRCs, where a lack of resources is placing enormous pressure on governments to accelerate the mass vaccination strategies. Accordingly, the establishment of new strategies that maximize the number of individuals who get vaccines without losing the efficacy of immune protection is urgently needed. The limited availability of authorized SARS-CoV-2 vaccines has led to widespread consideration of a single vaccine dose for people with past SARS-CoV-2 infection [110]. Recent evidence has suggested that one shot of SARS-CoV-2 vaccine could be all that is necessary to reach a high titer of antibodies for people who have had SARS-CoV-2 [111,112,113]. Accordingly, considering a one-shot vaccination program for SARS-CoV-2 previously infected individuals in African countries could have both economic and public health benefits [114]. To do this, a prevaccination screening approach would be used, in which individuals with evidence of a previous SARS-CoV-2 infection would be vaccinated with a single dose. Practically, implementing such a program necessitates the use of an antibody test to measure anti-SARS-CoV-2 antibodies and determine serostatus [115]. Ideally, such an antibody test should be both sensitive and specific in order to reduce false-positive and -negative results and to maximize the population-level benefit from the test and vaccination approach.

## 6. Resistance to SARS-CoV-2 Antibodies

Resistance to the SARS-CoV-2’s antibody is now a living fact due to the introduction of new virus variants, and this places an enormous load on the vaccination process. As such, recent vaccines have been developed to contradict the virus that was first discovered in late 2019 in Wuhan, China [116]. However, emerged variants such as the South African-Beta (B.1.351) and the UK-Alpha (B.1.1.7) have demonstrated extensive mutations in their S proteins, and these variants have been demonstrated to be highly contiguous [117]. Indeed, almost all monoclonal antibodies directed against the S protein’s N-terminal domain failed to recognize Alpha, although antibodies directed against the receptor-binding region were more effective [118]. Nonetheless, the variation exhibited reduced affinity for plasma from patients who have recovered from SARS-CoV-2 or sera from those who have been immunized against SARS-CoV-2. Both monoclonal antibodies directed against the N-terminal domain and several separate monoclonal antibodies directed against the receptor-binding motif are very ineffective against the mutated Beta, and this resistance might have been acquired from the mutated E484K substitution. Furthermore, Beta is much more resistant to neutralization by convalescent plasma (9.4-fold) and serum (10.3–12.4-fold) from BNT162b2-immunized individuals [118]. Furthermore, the Beta SARS-CoV-2 variants, which had various changes in their S proteins, were resistant to 17 neutralizing monoclonal antibodies as well as sera from convalescent patients and vaccinated mice, which were unable to neutralize the variants [119].

Subsequent studies indicated that certain Omicron variants are now resistant to antibodies elicited by vaccine doses. Remarkably, Omicron S proteins evaded blockage by antibodies obtained from persons inoculated with the BioNTech-Pfizer vaccine (BNT162b2) or convalescent patients with 12- to 44-fold more efficiency than the Delta (B.1.617.2) variant S protein [120]. However, heterologous doses of ChAdOx1 (Astra Zeneca-Oxford) and BNT162b2 or three doses of BNT162b2 caused significant antibody neutralization; however, the virus’s S proteins can still avoid neutralization more effectively than the B.1.617.2 S proteins [120]. More specifically, this resistance could be attributed to the reason that the N-terminal domain (NTD; 11 mutations with 6 deletions, 1 insertion, and the unique mutations N211Δ and ins214EPE) and RBD (two unique mutations: T547K and P681H) of the Omicron variant is highly mutated (Figure 1) [120]. These results show that the Omicron variant might be immune to the vast majority of the currently available antibody treatments. It also appears that the double-BNT162b2 immunization might not be sufficient to protect against the most virulent type. Supporting this notion, the antigenic characterization of new Omicron sublineages supports this notion, as polyclonal sera from infected patients with wild-type SARS-CoV-2 or receivers of existing mRNA vaccines showed a significant loss in neutralizing affectivity, both against BA.1 + R346K and BA.2, with declines similar to those previously noted for BA.1. In addition, BA.2 was shown to be refractory to 17 of the 19 monoclonal antibodies used in this investigation [121]. The BA.1 and BA.2 Omicron variants were also shown to be resistant to Regeneron (REGN10933 and REGN10987) and Lilly (LY-CoV555 and LY-CoV016) antibodies, but Vir-7831 and a combination of AstraZeneca monoclonal antibodies (AZD8895 and AZD1061) dramatically reduced neutralizing titer (Figure 1). In contrast, the LY-CoV1404 monoclonal antibody proved efficient in neutralizing both Omicron forms [122]. In addition, 18 antibodies lost their ability to neutralize Omicron sublineages BA 2.12.1 and BA.4/5. Three class 3 RBD antibodies (Brii-198, REGN10987, and COV2-2130) were inactive or impaired against BA. BA.4/5 showed poor neutralizing resistance to two class 3 RBD antibodies (REGN10987 and COV2-2130) and elevated neutralizing resistance to two class 2 RBD antibodies (ZCB11 and COV2-2196). These variations imply that, whereas BA.4/5 mutations facilitate the avoidance of both class 2 and class 3 RBD antibodies, BA.2.12.1 mutations facilitate the avoidance of class 3 RBD antibodies [123]. However, RBD CAB-A17, COV2-2130, 2–7, and LY-COV1404 antibodies showed neutralizing activity in vitro against both BA.2.12.1 and BA.4/5, with a half-maximum inhibitory concentration (IC50) less than 0.1 μg mL [123]. Although, the LY-CoV1404 (bebtelovimab) antibodies effectively neutralized the SARS-CoV-2 B.1.1.7, B.1.351, and B.1.617.2 variants, it also shows substantial neutralizing activity against several variants in pseudovirus neutralization reports, including B.1.1.7, B.1.351, B.1.617.2, B.1.427/B.1.429, P.1, B.1.526, B.1.1.529, and the BA.2 subvariant. It also interacts with spike proteins with RBD mutations, notably K417N, L452R, E484K, as well as N501Y [124].

Furthermore, the antibodies ReGN10933, REGN10987, and JS016 and serum from vaccinated participants neutralized the S protein of numerous wild-type variations, including Alpha, Beta, Gamma, and Delta. Regn10987 antibodies, for example, were the most effective in neutralizing the aforementioned variations. In addition, the Regn10933 and JS016 antibodies were both effective, although their reactions to the S protein were quite different [125]. These two antibodies, however, were unable to neutralize the Beta version. Unexpectedly, none of these antibodies neutralized the Omicron variant [125]. This result implies that these antibodies cannot be used to combat the ongoing Omicron variant pandemic or any developing variation.

Despite immunocompromised individuals potentially being more susceptible to SARS-CoV-2 viruses with unusual manifestations, prolonged immunosuppression treatment may provide some defense from severe COVID-19 disease consequences. However, the possibility of immunocompromised people acquiring abnormally severe COVID-19 is currently elusive [126]. Recent evidence shows that immunocompromised people may benefit greatly from convalescent plasma therapy [127,128,129,130], and that mutations and newly emerged virus strains will lead to more severe complications in the group. New variants, however, can appear in this patient population as a result of the selection pressure brought on by a severe viral infection [131]. The majority of people who have severe SARS-CoV-2 have immune system issues, thus the virus may live on for a very long period. Patients with impaired immune systems have been demonstrated to have varying SARS-CoV-2 evolution patterns [131]. How selection forces and evolutionary processes interact during chronic infection is an issue that has yet to be resolved. With that in mind, due to mutations (Q493K_RBD_, Q493K/R_RBD_) in the S protein in immunocompromised individuals, antibodies that were recovered from a healthy COVID-19 convalescent donor were shown to be ineffective in providing protection against SARS-CoV-2. In particular, the Q493K_RBD_ mutation 15-fold reduced the effectiveness of REGN10933 pseudotype neutralization, but the Q493K/R_RBD_ mutations almost completely imparted resistance to healthy COVID-19 convalescent donor IgG [132]. Additionally, research on SARS-CoV-2-infected immunocompromised patients revealed that convalescent plasma treatment may have contributed to the formation of the novel SARS-CoV-2 variants. In actuality, this novel type seemed to develop treatment resistance. In one experiment, repetitive passage of the virus, together with convalescent plasma, selected for antibody-resistant SARS-CoV-2 variants, including the E484K mutation which is linked to vaccination tolerance [133]. As a result, SARS-CoV-2 infection should be avoided in immunocompromised individuals because of their increased risk of COVID-19 consequences and the potential for the virus to grow and mutate. The link between immunosuppression and the production of more transmissible or more pathogenic SARS-CoV-2 variants demands further elucidation and mitigation measures since a sizable population of individuals worldwide suffers from inherent or induced immunosuppression [134].

## 7. Limitations of Antibody Tests

Although antibody tests are useful in COVID-19 case management and in vaccinations effectiveness, a number of drawbacks occur. The significant limitation is that antibodies may be present at undetectable levels in early days, thereby influencing the potency of any serodiagnostic tests and their effectiveness for diagnosing SARS-CoV-2 infection [135,136]. In this situation, a false-negative serological result from individuals with replicating and shedding viruses can have serious public health consequences [68]. Another limitation is the unknown duration at which IgM or IgG antibodies remain detectable after the virus has been cleared from the body. Furthermore, variations in antigens and methodologies used in IgM and IgG detection kits are essential, and they affect the sensitivity and specificity of the tests [137]. Unfortunately, the inaccuracy of antibody tests is unavoidable and will inevitably lead to false-positive or -negative results and disease misclassifications, particularly if these tests are not properly conducted and interpreted [138]. Moreover, the proven cross-reactivity of SARS-CoV-2 antibody tests with other coronaviruses is difficult to avoid [73]. Indeed, increasing the levels and duration of antibodies after SARS-CoV-2 vaccination to provide full protection and diagnostic opportunity against the current variants and those that can emerge in the future is still the primary objective in the upcoming studies.

## 8. Conclusions

In conclusion, the continuous spread of SARS-CoV-2 all over the world necessitates the utilization of anti-SARS-CoV-2 antibodies as efficient preventive and therapeutic agents to prevent, treat, and control the virus’s spread. For diagnostic purposes, antibodies could represent promising candidates and their potential role in the ongoing SARS-CoV-2 pandemic has generated considerable interest. Strong evidence suggests that it can be utilized for the routine diagnosis and clinical management of COVID-19. It can also be beneficial for the understanding of the spread of the virus amongst populations. Antibody testing helps epidemiologists to obtain seroprevalence data that can be used to estimate the proportion of infected people in the population. Furthermore, antibody testing will allow research scientists and vaccine developers to study the immune response during natural infection with SARS-CoV-2, as well as to evaluate vaccine efficacy in clinical trials. Indeed, understanding the humoral immune responses against SARS-CoV-2 and how antibodies are generated is vital for the development of therapeutic and vaccination options. However, more research is needed to determine the modulation of innate and vaccine-induced SARS-CoV-2 immunogenicity.

Due to the vital effect that antibodies play in the immune response to SARS-CoV-2, antibody cocktail treatments and potential vaccines may be utilized. These must be modified for effective protection against both the current and emerging SARS-CoV-2 types. Even though these techniques are being developed, heterologous or booster vaccines might help reduce the effect of any novel SARS-CoV-2 variations on population safety; standard control measures such as face masks and social seclusion should still be enforced.

## Figures and Tables

**Figure 1 vaccines-10-01346-f001:**
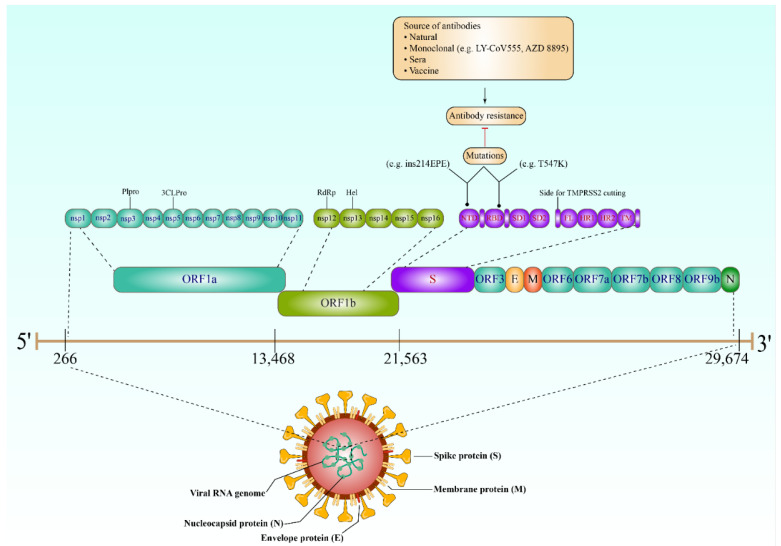
The SARS-CoV-2 genome, expected mutations (antibody resistance), and the structural proteins. The genome virus contains ORF1a (nsp1, 2, 3 (Plpro), 4, 5 (3CLPro), 6, 7, 8, 9, 10, and 11), ORF1b (nsp12 (RdRp), 13 (Hel), 14, 15, and 16), S (NTD, RBD, SD1/2, FL, HR1/2, and TM), ORF3, E, M, ORF6, ORF7a/b, ORF8, ORF9, and N, respectively. The complete virus also contains four structural proteins: Spike (S), nucleocapsid (N), membrane (M), and envelope (E) proteins, in addition to the viral RNA genome. Interestingly, mutations (e.g., T547 and ins214EPE) in the viral S gene (which produces the spike proteins, specifically NTD and RBD) have made the SARS-CoV-2 highly resistant to a wide range of antibodies, including monoclonal, those from natural infection or sera, and, finally, those produced after vaccination. RdRp, RNA-dependent RNA polymerase; RBD, receptor-binding domain; Plpro, papain-like protease; NTD, N-terminal domain; 3CLPro, 3C-like proteinase; SD1, subdomain 1, SD2 subdomain 2; Hel, helicase; HR1, heptad repeat 1; HR2, heptad repeat 2; TM, transmembrane domain; FL, fusion loop.

**Figure 2 vaccines-10-01346-f002:**
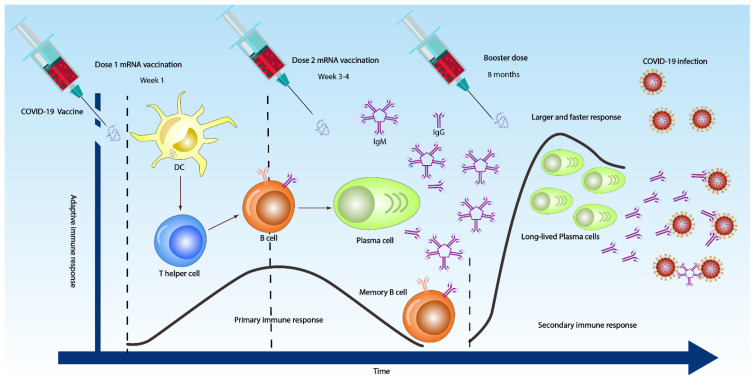
Overall antibody responses to 3 COVID-19 doses. After the first dose (sometimes IgM and IgG are detected as early as day 4 or 5 in week one), the dendritic cell (DC) in the body activates both T cells to differentiate into T helper 2 (Th2) cells and naïve B cell to differentiate into plasma cells. After the second dose at 3–4 weeks, the immune response becomes stronger and faster (primary immune response), producing high IgM and low IgG antibodies. However, booster vaccine injection might induce a very strong and vigorous immune response, with long-lived plasma cells producing low IgM and high IgG antibodies that may effectively opsonize, neutralizing the SARS-CoV-2 viruses (secondary immune response).

**Table 1 vaccines-10-01346-t001:** Characteristic data of different COVID-19 serological assays that employ S and/or M proteins.

SARS-CoV-2 Antigen (s)	Manufacturer	Test	Type of Test	Specimen	Target Antibody	Ref
S protein	Babson Diagnostics, Inc.	Babson Diagnostics aC19G1	CLIA	Serum and Plasma	IgG	[37]
	Xiamen Biotime Biotechnology Co., Ltd.	BIOTIME SARS-CoV-2 IgG/IgM Rapid Qualitative Test	RDT	Serum, Plasma, and Whole Blood	IgG and IgM	[38]
	Beijing Wantai Biological Pharmacy Enterprise Co., Ltd.	WANTAI SARS-CoV-2 Ab Rapid Test	RDT	Serum, Plasma, and Whole Blood	Pan-Ig	[39]
	Hangzhou Biotest Biotech	RightSign COVID-19 IgG/IgM Rapid Test Cassette	RDT	Serum, Plasma, and Whole Blood	IgG and IgM	[40]
	Healgen	COVID-19 IgG/IgM Rapid Test Cassette	RDT	Serum, Plasma, and Whole Blood	IgG and IgM	[41]
	Siemens Healthcare Diagnostics Inc.	ADVIA Centaur SARS-CoV-2 IgG (sCOVG)	CLIA	Serum and Plasma	IgG	[42]
	Diabetomics, Inc.	CovAb SARS-CoV-2 Ab Test	RDT	Oral fluid	IgG, IgA, and IgM	[43]
	NOWDiagnostics, Inc.	ADEXUSDx^®^ COVID-19 Test	RDT	Serum and Plasma	IgG	[44]
N protein	Abbott Laboratories Inc.	ARCHITECT SARS-CoV-2 IgG	CMIA	Serum and Plasma	IgG	[45]
	Roche Diagnostics, Inc.	Elecsys Anti-SARS-CoV-2	ECLIA	Serum and Plasma	Pan-Ig	[46]
	LumiraDx UK Ltd.	LumiraDx SARS-CoV-2 Ab Test	CLIA	Nasal Swab and Nasopharyngeal Swab	IgM and IgG	[47]
S + N proteins	Cellex Inc.	qSARS-CoV-2 IgG/IgM	RDT	Serum, Plasma, and Whole blood	IgG and IgM	[48]
	Access Bio, Inc.	CareStart COVID-19 IgM/IgG	RDT	Serum, Plasma, and Whole blood	IgG and IgM	[49]
	Bio-Rad Laboratories, Inc.	Platelia SARS-CoV-2 Total Ab	ELISA	Serum and Plasma	Pan-Ig	[50]
	Assure Tech. (Hangzhou Co., Ltd.)	Assure COVID-19 IgG/IgM Rapid Test Device	RDT	Serum, Plasma, and Whole Blood	IgG and IgM	[51]
	Sugentech, Inc.	SGTi-flex COVID-19 IgG	RDT	Serum, Plasma, and Whole Blood	IgG	[52]
	Inova Diagnostics, Inc.	QUANTA Flash SARS-CoV-2 IgG	CLIA	Serum and Plasma	IgG	[53]
	ACON Laboratories, Inc.	ACON Laboratories ACON SARS-CoV-2 IgG/IgM Rapid Test	RDT	Serum, Plasma, and Whole Blood	IgM and IgG	[54]
	Bio-Rad Laboratories	BioPlex 2200 SARS-CoV-2 IgG	CLIA	Serum and Plasma	IgG	[55]

Combined (Co), hours (H), receptor-binding domain (RBD), spike (S), spike 1 (S1), spike 2 (S2), and nucleocapsid proteins (N), chemiluminescent immunoassay (CLIA), rapid diagnostic test (RDT), chemiluminescent microparticle immunoassay (CMIA), electrochemiluminescence immunoassay (ECLIA), minute (min), enzyme-linked immunosorbent assay (ELISA), references (Ref).

## Data Availability

The data supporting this study’s findings are available from the corresponding author upon reasonable request.

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
