# Peer review of "The SARS-CoV-2 Antibodies, Their Diagnostic Utility, and Their Potential for Vaccine Development"

_vaccines, 2022, doi:10.3390/vaccines10081346_

Round 1

Reviewer 1 Report

Excellent review! I like very much the model.

Author Response

Response to Reviewer 1 Comments

Excellent review! I like very much the model.

Many thanks for your kind words and positive feedback

Reviewer 2 Report

Authors of this review initiated the abstract and introduction by touching upon limitations of diagnosis using PCR method to propose this can be replaced with antibody testing in resource limited countries. Later a big limitation of antibody testing in diagnosis is reported that it is only after 2-3 weeks after infection when antibodies become detectable -- this clearly go pass the infection period that lasts for for 5-14 days in general population leading to using antibody tests for diagnosis impossible.

Other aspects of utilizing antibody testing for true proportion of population previously infected and trying to find neutralizing antibodies can be the main target in this review. Here a full scale review of comparing all data on existing antibodies is required to make this review work more effective and useful. These two aspects require review of some data where antibodies for used for seroprevalence and detection of neutralizing antibodies.

At present only pointers to other papers are mentioned without taking any examples of neutralizing antibodies, for example antibodies from Omicron infection are able to neutralize all previous variants but not Omicron, see a recent work showing tests for neutralizing antibodies, https://www.nature.com/articles/s41467-022-31300-9.

There is no mention in the review for resistance caused to the SARS-CoV-2 when sera from recovered patients were used e.g. in the immunocompromised patients leading to further mutations in the virus with potential to create new variants.

It is recommended to revise this work considering the following main points:

1. Include appropriate data and figures showing comparison of antigens used for detection of infection, and how often these antigens became undetectable due to mutations and arrival of new variants.

2. Include appropriate data and figures on detection regarding neutralizing antibodies and how often these antibodies became ineffective due to mutations and arrival of new variants.

3. Include Impact of using sera from recovered patients in immunocompromised patients where further mutations were detected due to persistent infection and in response to treatment with sera containing antibodies from recovered patients.

4. As the viruses mutates further new variants arise, can we assess based on existing studies which show any neutralizing antibodies which are effective in neutralization of existing and future variants?

Answering such interesting questions based on existing studies and to include your assessment on where to fill the knowledge gaps for future studies will make your this work more impactful and useful.

Author Response

We would like to thank the academic editor and the reviewers for taking out their precious time to review this manuscript and give us their comments. We would like to explicitly state that we agree with all the comments as these helped us improve the quality of our paper. We have made a conscious effort to answer all the remarks in the paper as advised by the reviewers and track changes made for their convenience. Kindly consider these and excuse us for any lapse on our part.

Reviewer 3 Report

Although the authors try to address important and unresolved issues on the impact of laboratory detection of antibodies against SARS-CoV2, they do not succeed in presenting current data in a clear and educative way. Data on laboratory methods and clinicaly relevant  detection of IgM, IgG and IgA antibiodies against S protein (or N protein) of SARS-CoV2  is only generaly reported.The eview is full of redundancy of the main messages which are repeated across the text multipe times without giving enough evidence to support the central idea of this review;. This cannot be done only by reporting references without giving a concise and critical view of the studies. Moreove, thorough reviews on the uselfuness of the detection of antibodies in COVID-19 pandemic  have already existed (e g Chen Z et al,Clinical Reviews in Allergy & Immunology 2022). The authors  do not precise the population that can benefit from a large detection of antibody  tests for diagnostic purposes: are population in countries with no access or/and  a very low vaccination rate? Is this method cheaper and easily applicable compared to other standard- of- care methods? What is the most popular and reliable laboratory method of detection? (ELISA? other?). For all the above reasons, this review cannot be published under the present form as it do not add to the current knowledge on the field.

Author Response

(The authors gave the same response as above.)

Reviewer 4 Report

In this review, Hajissa et al., summarized about the COVID-19 antibodies. They introduced its diagnostic utility and potential for vaccine development. Overall this review explains antibody usage in whole area and should be good for readers. However, the authors should explain more about variants and need to improve Figure. In addition, there some typo and grammar errors. Please carefully check them.

Major

1.     Keywords: “antibody” and “antibodies” are same. Please delete one of them.

2.     Line 58 (As of December 12), can the authors update this information?

3.     The authors should describe more about the resistance of SARS-CoV-2 variants by vaccination elicited antibodies.

4.     The authors cited their Figure1 (line 89 and line 181 and line 195). But it looks like the sentence does not match the sentence. For example, there are no Omicron and IgA and day4 or 5 in the Figure. The authors should improve Figure 1. They need to at least add these information in the Figure or update everything.

5.     Line127-128: As far as I know, there are some B cell difference between severe and mild symptoms. Severe infection induces a robust extrafollicular response but an impaired germinal centre B cell response. However mild symptom induces both an extrafollicular response and GC B cell response. The authors should describe more about the B cells difference between severe and mild symptoms.

6.     Line 208: Please cite references.

7.     Line307-308: the authors should explain more about variants.

8.     Line 333-334: the authors should explain about TMPRSS2.

9.     Line 348: Please cite references.

10.  Line 356: “neutralization of viral infectivity” is not correct definition. Please fix it to “neutralization of virus” or “neutralization of viral spike”. 

11.  The authors might add additional Figure(s) to summarize their review. It might be better for readers.

Minor

1.     qRT-PCR should be RT-qPCR. Please fix them in the manuscript.

2.     Line135: COVID-19 viruses is not correct. Please fix it to SARS-CoV-2/.

3.     In the figure 1, please change “COVID-19 infection” to SARS-CoV-2 infection.

4.     Typo and grammar error

1.     line66: “s-“ is wrong

2.     line94: “SARS-COV2” should “SARS-CoV-2”

3.     line99: delete “that”

4.     line100: delete “t”

5.     line112: “stains” should be “strains”

6.     line131: ”T 2 helper” should be “T-helper 2 (Th2) cells” or “type 2 helper (Th2)”

7.     line132: ”into” is typo

8.     line204: ”rRT-PCR” should be RT-qPCR

9.     line227: ”antiby” and “utilised” are typo.

10.  line228: “RT-PCR” should be “RT-qPCR.

11.  line236: “immunisiation” is typo

12.  line351: delete “-“

13.  line364: “also” is typo

14.  line405: “furthermore” should start from capital letter

Author Response

(The authors gave the same response as above.)

Round 2

Reviewer 2 Report

Concerns raised in the first review are addressed to some extent, however a revision of English usage is required, it is suggested to seek help from a professional English language editing service.

Some example where such problems needs to be addressed:

section "6. Resistance of SARS-CoV-2 antibodies", 

Is this about resistance to SARS-CoV-2 antibodies or resistance of SARS-CoV-2 antibodies.

Also, the next line "Resistance to the SARS-CoV-2's antibody is now a live actuality due to the introduction of new virus variants, and this places an enormous load on the vaccination process."

This sentence is not clear.

Next "However, emerged variants such as the South African-B.1.351 and the UK-B.1.1.7 have demonstrated extensive mutations in their S proteins and these variants are highly contiguous [116]. B.1.1.7 is resistant to the vast majority of monoclonal antibodies that target the N-terminal domain of the S protein, but less resistant to antibodies that target the receptor-binding region [117] "

These variants are no more detected so may be referring to these can be done in past tense.

The following sentence is also not clear, what are probable vaccines?

Due to the vital effect that antibodies play in the immune response to SARS-CoV-2, antibody cocktail treatments and probable vaccines may be utilized.

Please throughly check all other sections of the paper to improve the clarity of the context.

Author Response

We appreciate the editor and reviewers for taking the time to assess our work and provide helpful feedback and legitimate criticism. We found them extremely useful in detecting flaws and ambiguities in our paper and carefully analyzed them. Accordingly, the manuscript has been revised. We believe that by doing so, the revised manuscript has been significantly enhanced. The following responses have been prepared to address all of the reviewers’ comments point-by-point.

Reviewer 4 Report

Hajissa et al., summarized about the COVID-19 antibodies in this review. Based on my comments, the authors improved the manuscript. Now the manuscript became better, however they need to improve the paper a little bit. The following is my minor comments.

Minor

1.     Line 53, 54: “COVID-19 infection” is not correct. Please fix them to “SARS-CoV-2 infection” or “COVID-19”. Also, please check through the manuscript.

2.     Line 387: The reference is cited in wrong line. Please fix it.

3.     For example, line 454 and 455 and 474, the authors use B.1.1.7 and B.1.351 and B.1.617.2 to show variants. However they need to use Alpha and Beta and Delta (must be same as line 494).

4.     Line 465: “501Y.V2 SARS-CoV-2” is Beta (B1.1351). please fix the name.

5.     In the section 6, please add the information about Omicron BA.3, 4, 5. Most monoclonal antibodies are resistant against BA.3, 4, 5, however LY-1404 still works against these variants. The authors need to add at least this information.

6.     Line 464:  Reference [117] tested the serum neutralizing activity from BNT162b2 immunized individuals (healthy donor). So “immunized COVID-19 patients” should be “BNT162b2 immunized individuals”

7.     Line 497: “Beat’ is typo?

8.     Line 503: “COVID1-9” is typo.

9.     Line 492: Regn10933 and Regn10977 should be ReGN10933 and REGN10987.Please fix them.

Author Response

(The authors gave the same response as above.)
